# How many submissions are needed to discover friendly suggested reviewers?

**Pedro Pessoa**[1,2], **Steve Pressé**[1,2,3]*

**1** Center for Biological Physics, Arizona State University, Tempe, AZ, United States of America, **2** Department of Physics, Arizona State University, Tempe, AZ, United States of America, **3** School of Molecular Sciences, Arizona State University, Tempe, AZ, United States of America

* spresse@asu.edu

## Abstract

It is common in scientific publishing to request from authors reviewer suggestions for their own manuscripts. The question then arises: How many submissions are needed to discover friendly suggested reviewers? To answer this question, as the data we would need is anonymized, we present an agent-based simulation of (single-blinded) peer review to generate synthetic data. We then use a Bayesian framework to classify suggested reviewers. To set a lower bound on the number of submissions possible, we create an optimistically simple model that should allow us to more readily deduce the degree of friendliness of the reviewer. Despite this model's optimistic conditions, we find that one would need hundreds of submissions to classify even a small reviewer subset. Thus, it is virtually unfeasible under realistic conditions. This ensures that the peer review system is sufficiently robust to allow authors to suggest their own reviewers.

## 1 Introduction

Peer review is the cornerstone of quality control of academic publishing. However, the daunting task of selecting appropriate reviewers [1, 2] relies in identifying at least two scholars, free of conflict of interest, who have: 1) the necessary expertise to judge the quality and perceived impact; and 2) the willingness to perform the work *pro bono*. On account of this, it is ever more common that journals request, and often require, authors to suggest candidate reviewers. That is, provide names and contact information of scholars the authors deem qualified to review.

It is natural to imagine, at first glance, that this incentivizes authors to submit "friendly" names, implying suggesting reviewers that they have reason to believe would be favorably inclined toward them. The fear of such peer review manipulation is potentiated by reports that author-suggested reviewers are more likely to recommend acceptance [3–10]. However, some of these same studies mention that the quality of reports of author-suggested reviewers does not differ from the ones of editor-suggested reviewers [3–5, 8, 9]. It is also reported that the difference in suggesting acceptance by author-suggested and editor-suggested reviewers is not significant when comparing reports of the same submission [7] nor is it observed to have an

**Data Availability Statement:** The code performing the simulation, inference, and generating figures is available on GitHub https://github.com/PessoaP/how_many_submissions.

**Funding:** This work is supported by funds from the National Institutes of Health (https://www.nih.gov/)

grant No. R01GM134426, R01GM130745, and the MIRA R35 entitled "Toward high spatiotemporal resolution models of single molecules for in vivo applications", all of which were awarded to SP. The funder did not play any role in the study design, data collection and analysis, decision to publish, or preparation of the manuscript.

**Competing interests:** The authors have declared that no competing interests exist.

effect in the article's acceptance [3, 7] and this discrepancy can even vanish entirely in some fields [11].

The question then naturally arises: can a scientist infer from their personal history of submissions which reviewers are likely to bias the decision in their favor? In what follows, we present an optimistic agent-based model that surely underestimates the number of submissions required to ascertain the friendliness of the reviewer with high confidence. What we find is that, due to multiple sources of uncertainty (*e.g.*, lack of knowledge as to which reviewer the editor selects), such an effort would require a number of submissions vastly exceeding the research output of all but the most productive scientists. That is, hundreds and sometimes thousands of submissions.

As neither a manuscript's submission history, reviewers selected by the editor, nor suggested reviewers by the authors are publicly available, we adapt an agent-based simulation model [12–14], already used in generating simulated peer review data [14], and develop an inference strategy on this model's output to ask whether we can uncover favorably inclined reviewers. This fits into a larger effort to quantitatively study the dynamics of scientific interactions [15–18].

As we initially simulate the data, we intentionally make assumptions using agent-based models that would result in easy classification in order to obtain a lower bound on the number of submissions required to confidently classify reviewers. These assumptions read as follows:

i). For each submission, the author will always suggest a small number of reviewers (three, in our simulation) from a fixed and small (ten elements, in our simulation) pool of names.

ii). The editor will always select one of the reviewers suggested by the authors.

iii). The "friendliness" of any given reviewer remains the same for all subsequent submissions.

iv). Submissions from the same author all have the same overall quality.

Shortly we will lift the assumptions of this "cynical model" and introduce a "quality factor model" or simply, quality model. In particular, we will lift assumption iv). As we will see, lifting assumptions will only raise, often precipitously, the already unfeasible high lower bound on the number of submissions required to confidently classify reviewers and leverage this information to bias reports in their favor.

## 2 Methods

In order to set a lower bound on the number of submissions required to confidently classify reviewers, the present study focuses on a simplified peer review process characterized by three types of agents: the author(s), the editor, and the reviewers. Each submission is reviewed according to the following steps:

1. During submission, the author will send to the editor a list of suggested reviewers, $\mathcal{S}$. The suggested reviewers are chosen from a larger set of possible reviewers $\mathcal{R}$—such that $\mathcal{S}$ is a subset of $\mathcal{R}$.

2. The editor will select one reviewer, namely $r_1$, from $\mathcal{S}$ randomly with uniform probability.

3. The editor will also select a second reviewer, $r_2$, from a pool of reviewers considerably larger than $\mathcal{R}$ and representative of the scientific community.

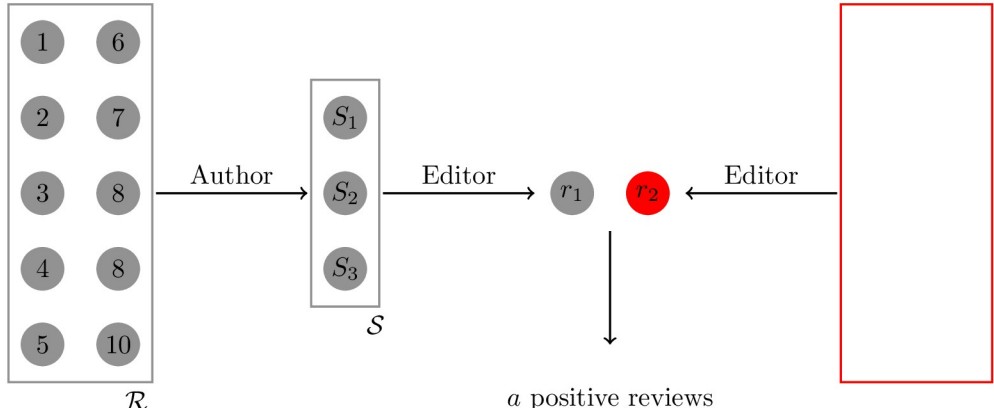

**Fig 1. Diagram presenting the simplified peer review process.** See the steps described in section 2 for definitions.

4. The reviewers will write single blind reports, that will be shared with the authors. These are classified as overall positive or negative, with $a$ being the number of positive reviews out of two reports.

A diagram of this idealized process is presented in Fig 1.

In the spirit of identifying a lower bound on submissions, we make the dramatic assumption that $r_1$ either belongs to friend or rival class while $r_2$ is otherwise neutral. Later we will devise a Bayesian inference strategy to achieve suggested reviewer ($r_1$) classification.

The procedure described in the bullet points above refers to a single submission. However, as our end goal is to determine how many submissions are necessary to classify reviewers, we must consider multiple submissions. For this reason, we represent a history of $M$, identical and independent, submissions using the index $\mu \in \{1, 2, \ldots, M\}$, such that $\mathcal{S}_\mu$ and $a_\mu$ are, respectively, the set of suggested reviewers and positive reviews accrued for the $\mu$-th submission. Naturally, reviewer reports are fully written letters rather then binary positive or negative answers. However, introducing a more complete spectrum on the positiveness of a report would certainly introduce further uncertainty on the reviewers class and only increase the lower bound necessary to classify reviewers.

Now that we have qualitatively described our agent-based model, we provide a detailed mathematical formulation of the simulation and inference.

## 2.1 Mathematical formulation

Here each element of $\mathcal{R}$ is a reviewer. We denote $x_i$ the state of each reviewer as belonging to one of two classes: either $x_i = friend$ or $x_i = rival$. The method can immediately be generalized to accommodate the addition of a third (neutral) class. Put differently, each suggested reviewer is treated as a Categorical random variable realized to either friend or rival. Collecting all states as a sequence, we write $x = \left[x_1, x_2, \ldots, x_{|\mathcal{R}|}\right]$ with $|\mathcal{R}|$ understood as the cardinality of $\mathcal{R}$. For two classes, we have $2^{|\mathcal{R}|}$ allowed configurations of $x$. It is convenient to index configurations with a $j$ superscript where $j \in \{1, 2, \ldots, 2^{|\mathcal{R}|}\}$ for which $x^j = \left[x_1^j, x_2^j, \ldots, x_{|\mathcal{R}|}^j\right]$. For sake of clarity alone, we provide a concrete example enumerating all configurations for two possible suggested reviewers in Table 1.

**Table 1. Example of the construction and enumeration of the possible configurations, $x_j$, for a set of two possible suggested reviewers ($|\mathcal{R}| = 2$).** As described in the first paragraph of section 2.1.

| j | $x^j$ | = [ | $x_1^j$ | $x_2^j$ | ] |
|---|---|---|---|---|---|
| 1 | $x^1$ | = [ | rival, | rival | ] |
| 2 | $x^2$ | = [ | rival, | friend | ] |
| 3 | $x^3$ | = [ | friend, | rival | ] |
| 4 | $x^4$ | = [ | friend, | friend | ] |

We will now use Bayesian inference to determine the probability we assign to each configuration. That is, to compute posterior probabilities, $P(x^j|\{a_\mu\}, \{\mathcal{S}_\mu\})$, over each $x^j$ given the set of positive reports $\{a_\mu\}$ received after suggesting a subset $\{\mathcal{S}_\mu\}$ of reviewers. In the present article, we will study two models for reviewer behavior.

The first is the, simpler, cynical model where the friend writes a positive review with unit probability and, by contradistinction, the rival writes a positive review with null probability. The reviewer not selected from the author's list, $r_2$, will write a positive review with probability ½. In this iteration of the model it should be the easiest (*i.e.*, quickest in terms of number of submissions) to sharpen our posterior and classify reviewers.

The second model is the quality model that introduces a new layer of stochasticity. Here, a submission is associated a quality factor $q \in (0, 1)$ reflecting the quality of each submission. In this model an unbiased reviewer ($r_2$) would write a positive review with probability $q$. By contrast, rivals and friends will "double guess" their own judgment of the article implying that they will evaluate the submitted article twice independently. A rival will only suggest acceptance if they deem the submission worthy of publication in both assessments, meaning a rival will write a positive review with probability $q^2$. Analogously, a friend will reject if they "reject twice", hence they write a negative review with probability $(1 - q)^2$ or, equivalently, a positive review with probability $1 - (1 - q)^2 = q(2 - q)$. A summary of these probabilities is presented in Table 2. As done with $a_\mu$ and $\mathcal{S}_\mu$, we index the quality factor of the $\mu$-th submission as $q_\mu$.

Not all authors, naturally, have distributions over $q$ centered at the same value. It is therefore of interest to compute the effect on the lower bound of submission needed (*i.e.*, how quickly our posterior sharpens around the ground truth) for different distributions over $q$ centered at the extremes (average high or average low quality) in addition to middle-of-the-road distributions centered at $q = $ ½. As we will see, middle-of-the-road distributions allow for more rapid posterior sharpening. Notwithstanding this paltry incentive to write middle-of-the-road papers, we will see that the lower bound on the number of submissions remains unfeasibly high. Even for this idealized scenario.

## 2.2 Simulation

Following the steps described at the beginning of Section 2, the first step of the simulation involves editorial selection from the list of suggested reviewers with $\binom{|\mathcal{R}|}{|\mathcal{S}_\mu|}$ possible sets of

**Table 2. Probabilities for reviewers of each class to write a positive report (accept) or a negative report (reject) according to the quality model when reviewing a paper of quality factor $q$.**

| | accept | reject |
|---|---|---|
| $r_2$ | $q$ | $1 - q$ |
| $r_1$ is a *rival* | $q^2$ | $1 - q^2$ |
| $r_1$ is a *friend* | $1 - (1 - q)^2 = q(2 - q)$ | $(1 - q)^2$ |

suggested reviewers possible, or 120 given our simulation parameters ($|\mathcal{S}_\mu| = 3$ and $|\mathcal{R}| = 10$ for all $\mu$). Each $\mathcal{S}_\mu$ for any $\mu$ is independently sampled with uniform probability. We show the effects of changing the number of suggested reviewers (four and five reviewers per submission) in S1 File.

To start our simulation, we must initialize the ground truth configuration (the identity of $x$). Initially, we set an equal number of friends and rivals though we generalize to two other cases (seven and nine friends) in the S2 File.

The subsequent steps of the simulation of the data (steps 2–3) are straightforward. Step 4 for the cynical model is equally straightforward (and deterministic in $r_1$): a positive review is returned if $r_{1_\mu}$ is a friend, a negative review s returned otherwise, while $r_2$ writes a positive review with probability ½. Further mathematical simulation details are found in S3 File.

For the quality model, to each submission ($\mu$) is associated a quality factor $q_\mu \in (0, 1)$. As is usual for a variable bounded by the interval (0, 1), we we take $q_\mu$ as a Beta random variable such that

$$P(q_\mu) = \frac{q_\mu^{\alpha-1}(1 - q_\mu)^{\beta-1}}{B(\alpha, \beta)} \tag{1}$$

where $B(\alpha, \beta) = \frac{\Gamma(\alpha)\Gamma(\beta)}{\Gamma(\alpha+\beta)}$ where $\Gamma$ being the Euler's gamma function. Again, in an effort to compute a lower bound alone on the number submissions required, we assume that all $q_\mu$ are sampled from the same, stationary, distribution with constant $\alpha = 12$ and $\beta = 12$ for now (middle-of-the-road quality distribution) for which the mean $\langle q \rangle$ = ½ and the variance is $\sigma_q$ =.01.

In reality, it is conceivable that one's quality factor distribution shifts to the right with experience. It is also conceivable that a prolific researcher would have its quality factor shift to the left as they start venturing into new fields. This effect only makes it harder to assess which reviewer is friendly and further raises the lower bound required on the number of submissions. In any case, in the S4 File, we consider different quality distributions (both high and low). Foreshadowing the conclusions, it may be intuitive to see that very high or very low quality factors result in less information gathered per reviewer report. That is, we learn best the class to which reviewers belong by sampling quality factors around ½. Not by constant rejection or acceptance.

Thus, with each sampled $q_\mu$, step 4) of the quality model is implemented by observing that reviewers write positive reviews according to the probabilities in Table 2. Further mathematical details of the quality model are relegated to S3 File.

Importantly, for the purposes of classifying which reviewers are friendly, it is not necessary to know whether the article is accepted by the editor, only the count of positive or negative reviews per submission.

## 2.3 Inference strategy

Inference consists of constructing the posterior $P(x^j|\{a_\mu\}, \{\mathcal{S}_\mu\})$ and drawing samples from it. To construct this posterior, we update the likelihood, $P(\{a_\mu\}|x^j, \{\mathcal{S}_\mu\})$, over all independent submission

$$P(\{a_\mu\}|x^j, \{\mathcal{S}_\mu\}) = \prod_\mu P(a_\mu|x^j, \mathcal{S}_\mu) \tag{2}$$

as follows

$$P(x^j|\{a_\mu\}, \{\mathcal{S}_\mu\}) = \frac{P(x^j|\{\mathcal{S}_\mu\})}{P(\{a_\mu\}|\{\mathcal{S}_\mu\})} \; P(\{a_\mu\}|x^j, \{\mathcal{S}_\mu\}) \; . \tag{3}$$

Since the number of configurations is finite, we may start by taking the prior as uniform over these countable options ($P(x^j|\{\mathcal{S}_\mu\}) = 2^{-|\mathcal{R}|}$). Keeping all dependency on $x^j$ explicit, we may write

$$P(x^j|\{a_\mu\}, \{\mathcal{S}_\mu\}) \propto P(\{a_\mu\}|x^j, \{\mathcal{S}_\mu\}) = \prod_\mu P(a_\mu|x^j, \mathcal{S}_\mu) \; . \tag{4}$$

We end with a note on the likelihood which we compute explicitly by treating $r_{1_\mu}$ as a latent variable over which we sum. That is,

$$P(a_\mu|x^j, \mathcal{S}_\mu) = \sum_{r_{1_\mu}} P(a_\mu|r_{1_\mu}) \; P(r_{1_\mu}|x^j, \mathcal{S}_\mu) \; . \tag{5}$$

In terms of the factors within the summation, $P(r_{1_\mu}|x^j, \mathcal{S}_\mu)$ follows from step 2). That is, if the editor selects $r_{1_\mu}$ with uniform probability from $\mathcal{S}_\mu$, the probability of selecting a $r_{1_\mu}$ from the class of friends is the ratio of friends, $f$, in $\mathcal{S}_\mu$ according to the configuration $x^j$. This can be written more rigorously as

$$P(r_{1_\mu} = friend \; |x^j, \mathcal{S}_\mu) = f(x^j, \mathcal{S}_\mu) \doteq \frac{1}{|\mathcal{S}|} \sum_{i \in \mathcal{S}} F(x_i^j), \tag{6}$$

where

$$F(x_i^j) = \begin{cases} 0 & \text{if} \quad x_i^j = rival \\ 1 & \text{if} \quad x_i^j = friend \end{cases} . \tag{7}$$

It follows that $P(r_{1_\mu} = rival \; |x^j, \mathcal{S}_\mu) = 1 - f(x^j, \mathcal{S}_\mu)$.

We now turn to the term $P(a_\mu|r_{1_\mu})$ within (5) computed differently within both the cynical and quality models.

**2.3.1 Inference in the cynical model.** Calculating $P(a_\mu|r_{1_\mu})$ for the cynical model is straightforward. That is, given that a friendly $r_1$ always writes a positive review and a rival $r_1$ always writes a negative one, and $r_2$ writes a positive review with probability ½, values for $P(a_\mu|r_{1_\mu})$ immediately follow as tabulated in Table 3. Eqs (3)–(7) and Table 3 summarize what is needed to perform Bayesian classification within the cynical model formulation.

**2.3.2 Inference in the quality model.** The major difference between inference in the quality and cynical models relies on the fact that the author will not have access to individual $q_\mu$'s. However, since we aim for a lower bound, we will proceed with the calculation under the assumption that while individual $q_\mu$'s are unknown the author knows the distribution from

**Table 3. Probabilities for the number of positive reports, $a_\mu$, in the cynical model, conditioned on the class of the suggested reviewer, $r_{1_\mu}$.**

| $P(a_\mu|r_{1_\mu})$ | $a_\mu = 0$ | $a_\mu = 1$ | $a_\mu = 2$ |
|---|---|---|---|
| $r_{1_\mu} = friend$ | 0 | ½ | ½ |
| $r_{1_\mu} = rival$ | ½ | ½ | 0 |

**Table 4. Probability for the number of positive reviews $a_\mu$ conditioned on the quality factor $q_\mu$ and the class of the reviewer $r_{1_\mu}$.**

| $P(a_\mu|q_\mu, r_{1_\mu})$ | $a_\mu = 0$ | $a_\mu = 1$ | $a_\mu = 2$ |
|---|---|---|---|
| $r_{1_\mu} = friend$ | $1 - 3q_\mu + 3q_\mu^2 - q_\mu^3$ | $3q_\mu - 5q_\mu^2 + 2q_\mu^3$ | $2q_\mu^2 - q_\mu^3$ |
| $r_{1_\mu} = rival$ | $1 - q_\mu - q_\mu^2 + q_\mu^3$ | $q_\mu + q_\mu^2 - 2q_\mu^3$ | $q_\mu^3$ |

which $q_\mu$ is sampled. If the author were uncertain of the distribution, this would add yet another layer of stochasticity and further raise the lower bound. From Table 2, it is straightforward to calculate the probability of each $a_\mu$ given $r_{1_\mu}$ and $q_\mu$ in the quality model. The result is found in Table 4.

Without access to $q_\mu$ in (5), we further need to marginalize $P(a_\mu|q_\mu, r_{1_\mu})$ over $q_\mu$ as follows

$$P(a_\mu|r_{1\mu}) = \int dq_\mu \; P(a_\mu, q_\mu|r_{1\mu}) = \int dq_\mu \; P(a_\mu|q_\mu, r_{1\mu})P(q_\mu) = \langle P(a_\mu|q_\mu, r_{1\mu})\rangle_{q_\mu} \; . \quad (8)$$

For example, if $q_\mu$ is sampled from a Beta distribution (1) with parameters $\alpha = \beta = 12$, as proposed in Section 2.2, marginalization (8) yields the values of $P(a_\mu|r_{1_\mu})$ shown in Table 5.

Inference occurs, otherwise, exactly as in the cynical model, thus summarized by Eqs (3)–(8), and Table 4.

# 3 Results

The previous section was focused on constructing the $2^{|\mathcal{R}|}$-dimensional posterior $P(x^j|\{a_\mu\}, \{\mathcal{S}_\mu\})$ otherwise difficult to visualize. Since our goal is to determine the number of submissions required to correctly classify suggested reviewers, we introduce metrics measuring how well the posterior classifies reviewers. Moreover, these metrics ought to be have an assigned value at each submission, and thus be a function of $m$ for each $m \in \{1, 2, \ldots, M\}$. Thus, for a fixed data set of $M$ submissions, we calculate each metric using the first $m$ submissions for all $m$.

Each metric is a stochastic function dependent on the dataset (decisions made by reviewers and quality factors sampled) inherited from the variation of the posterior with the data supplied. For this reason, we consider multiple metric realizations which allow us to compute their mean, median and 50% and 95% credible (or confidence) intervals. Borrowing language from dynamical systems, we refer to these realizations, up to the $m$th submission, as trajectories.

## 3.1 Metrics

The first metric, akin to a marginal decoder obtained for mixture models [19, 20], concerns itself with the probability for the class of one specific reviewer $i$. From the posterior over all

**Table 5. Probability for the number of positive reviews $a_\mu$ conditioned on the class of the reviewer $r_{1_\mu}$.** Calculated by marginalizing $q_\mu$ in Table 4, as described in (8), for $\alpha = \beta = 12$.

| $P(a_\mu|r_{1_\mu})$ | $a_\mu = 0$ | $a_\mu = 1$ | $a_\mu = 2$ |
|---|---|---|---|
| $r_{1_\mu} = rival$ | .38 | .48 | .14 |
| $r_{1_\mu} = friend$ | .14 | .48 | .38 |

configurations, we obtain probabilities over the reviewer $i$'s class through marginalization

$$
\begin{aligned}
P(x_i = friend \ |\{a_\mu\}, \{\mathcal{S}_\mu\}) \ &= \sum_j P(x_i^j = friend \ |\{a_\mu\}, \{\mathcal{S}_\mu\}) \\
&= \sum_j P(x^j|\{a_\mu\}, \{\mathcal{S}_\mu\}) \ F(x_i^j) \ ,
\end{aligned}
\tag{9}
$$

where $F$ was defined in (7). Equivalently, $P(x_i = rival) = 1 - P(x_i = friend)$.

Thus, the first metric is defined as the marginal probability of reviewer $i$ being a friend based on the results of $m$ papers where reviewer $i$ was suggested

$$
\rho_i(m) \doteq P(x_i = friend \ |\{a_\mu\}_{\mu=1:m}, \{\mathcal{S}_\mu\}_{\mu=1:m}) \ ,
\tag{10}
$$

where $\{a_\mu\}_{\mu=1:m}$ and $\{\mathcal{S}_\mu\}_{\mu=1:m}$ represents the subset of the $m$ first elements of $\{a_\mu\}$ and $\{\mathcal{S}_\mu\}$ respectively.

The second metric, a global metric, simply compares the maximum *a posteriori* (MAP) estimate for $x$ after $m$ submissions, $\bar{x}(m)$,

$$
\bar{x}(m) \doteq \arg\max_{x^j} \ P(x^j|\{a_\mu\}_{\mu=1:m}, \{\mathcal{S}_\mu\}_{\mu=1:m}) \ ,
\tag{11}
$$

and compares, element-wise, how $\bar{x}(m)$ differs from the ground truth.

The simulated MAP error, while less informative than considering the full posterior, serves as a estimate on the number of submissions necessary to estimate lower bounds (within tolerable error) to classify as a function of $m$ and the number of friends in the original pool of reviewers. More robustness analysis is performed in S2 File.

As a third metric, we look for a more general metric for how "well-classified" the reviewers are. Following the work of Shannon [21], we notice that entropy defined as

$$
S(m) \doteq -\sum_{x^j} P(x^j|\{a_\mu\}_{\mu=1:m}, \{\mathcal{S}_\mu\}_{\mu=1:m}) \ \log_2 P(x^j|\{a_\mu\}_{\mu=1:m}, \{\mathcal{S}_\mu\}_{\mu=1:m}) \ ,
\tag{12}
$$

measures, in rough terms, how many reviewers are left unclassified (Base 2 for the logarithm in (12) was chosen because we are dealing with binary classification.). A mock example on how entropy works for the classification of 2 reviewers is presented in Table 6. For more general insight on the role of entropy see *e.g.*, Refs. [22–24] and references therein.

**Table 6. Example for how entropy is to be interpreted. In this mock example for the classification of two reviewers, similar to Table 1, all probabilities over configurations (first column) are equally likely and thus the entropy is ascribed its maximal value of 2.** The second column shows a case where the first reviewer is not yet classified, but is considerably more likely to belong to one class, hence entropy takes on some value between 1 and 2. The third column contains an example where the first reviewer is fully identified, but the probability does not favor any classification for the second reviewer leading to the entropy value of 1. The fourth column has an example where the first reviewer is fully identified, but the probability favors one classification (rival) for the second reviewer, leading to the entropy value between 0 and 1. The last column contains an example where one configuration has probability 1, hence the reviewers are fully classified, leading to 0 entropy.

| $j$ | $x^j$ | $= [$ | $x_1^j$ | $x_2^j$ | $]$ | $P(x^j)$ | $P(x^j)$ | $P(x^j)$ | $P(x^j)$ | $P(x^j)$ |
|---|---|---|---|---|---|---|---|---|---|---|
| 1 | $x^1$ | $= [$ | rival, | rival | $]$ | ¼ | ⅛ | 0 | 0 | 0 |
| 2 | $x^2$ | $= [$ | rival, | friend | $]$ | ¼ | ⅛ | 0 | 0 | 0 |
| 3 | $x^3$ | $= [$ | friend, | rival | $]$ | ¼ | ⅜ | ½ | ⅞ | 1 |
| 4 | $x^4$ | $= [$ | friend, | friend | $]$ | ¼ | ⅜ | ½ | ⅛ | 0 |
| entropy | | $-\sum_j P(x^j) \log_2 P(x^j)$ | | | | 2 | $\approx 1.812$ | 1 | $\approx 0.5436$ | 0 |

The fourth, and final, metric is the third largest marginal posterior, or the posterior for the third reviewer most likely to be friendly,

$$T(m) \doteq \max_i^3 \rho_i(m) \ , \tag{13}$$

where $\max_i^n$ is the *n*-th biggest element in the set indexed by *i* and $\rho_i(m)$ is defined in (10). Unlike the first metric, which classifies each reviewer individually, and second and third metrics, which classify all reviewers in $\mathcal{R}$, this metric classifies a scenario where authors only seek to classify a minimum number of suggested reviewers ($|\mathcal{S}_\mu| = 3$ in our simulations). Therefore, whenever we present results for this fourth metric, we show how many publications are required in order to reach the 95% confidence level. Despite reaching this metric, it is possible to misclassify the third referee; details provided in S5 File. In the same Supporting Information we also explore the possibility that suggesting reviewers based on outcomes from prior optimization on previous submissions does not lead to significant reduction in submissions necessary to classify reviewers.

## 3.2 Cynical model results

The marginal probability (first metric) for the reviewer belonging to the friend class in the cynical model is shown in Fig 2. We interpret this result as indicating that one needs to suggest this reviewer in a little over than 75 submissions to strongly classify (marginal posterior exceeding 0.95 for one of the classes) this reviewer for the median case. Assuming this reviewer is picked uniformly from the author's pool of 10 reviewers then, on average, a total number of 250 submissions would be required.

By contrast, around 100 submissions suggesting this reviewer are necessary to weakly classify, meaning classify this reviewer using the class that has the highest marginal posterior and obtain the correct class within the 95% credible interval. In S2 File we see that if we have more friends in the ground truth configuration, friends are classified faster, but rivals are likely to be misclassified.

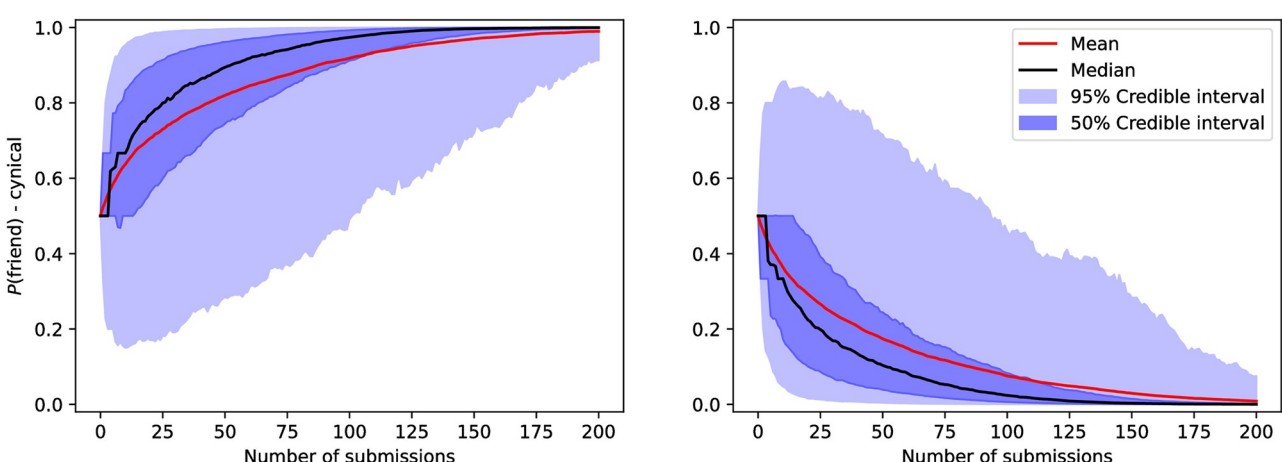

**Fig 2. Posterior marginal probability of a single reviewer's class—$\rho_i$ defined in (10)—as a function of the number of submissions where the reviewer was suggested.** The graph on the left corresponds to values of $\rho_i(m)$ for which the reviewer belongs to the friendly class in the simulation's ground truth, while the graph on the right corresponds to rivals in the simulation's ground truth. We observe that the median trajectory reaches a probability of .95 for the correct class after a little more than 75 submissions involving the suggested reviewer. Meanwhile, it takes between 100 to 125 for the class with the highest posterior to match ground truth within the 95% credible interval for submissions involving this reviewer.

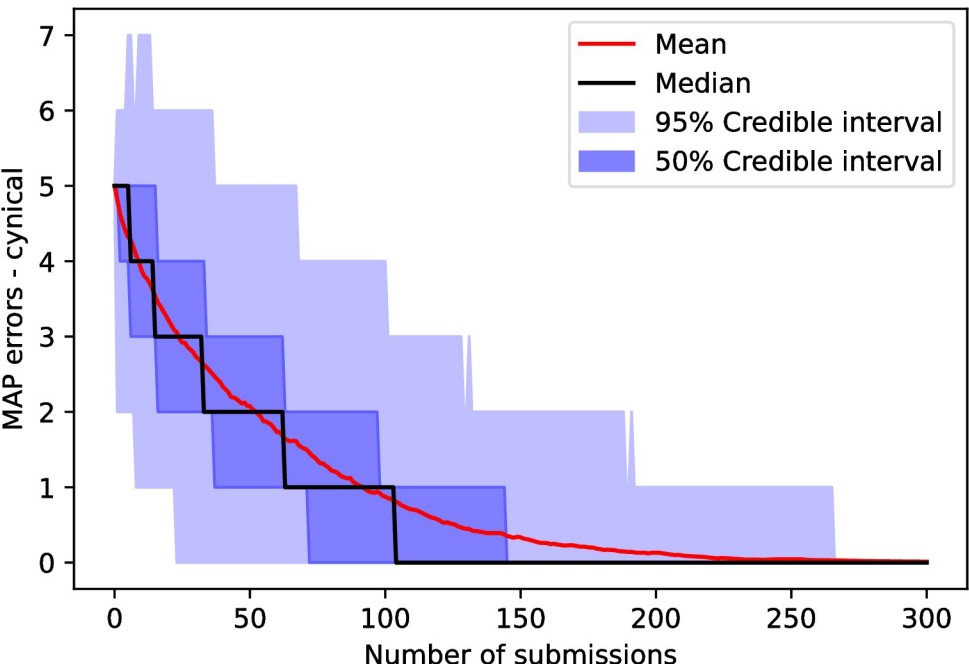

**Fig 3. Number of errors when using maximum *a posteriori* (MAP) classification, *i.e.*, the number of misclassifications appearing in the MAP configuration (11) when comparing to the simulation's ground truth as a function of the number of submissions in the cynical model.** We observe that the median trajectory finds the correct ground truth configuration using the MAP estimate after approximately 100 submissions, while it takes approximately 250 submissions to reach the correct configuration within the 95% credible interval.

The number of errors from the MAP (second metric) for the cynical model as a function of the number of submissions is shown in Fig 3. There, we can see that if we attempt to classify reviewers using the MAP, we would get the correct configuration, in the median case, after approximately 100 submissions. However, to guarantee one finds the correct configuration within the 95% confidence interval, it needs between 250 to 300 submissions.

The posterior entropy (third metric) for the cynical model as a function of the number of submissions is shown in Fig 4. In this case, we would need, in the median case, between 150 and 200 submissions to fully classify a set of 10 reviewers with 3 suggested per submission. In the S2 File, we see that the posterior entropy does not fall considerably faster (as compared to this case with 5 friends) with more friends in the ground truth.

Finally, we present the third largest marginal posterior as a function of the number of submissions in Fig 5. We observe that it takes approximately 80 submissions for the median trajectory to reach $T(m) = 0.95$. In the same figure, we also see that it takes on average 70 submissions to reach that confidence for all top 3 reviewers. In the S5 File, we show that if one stops classifying reviewers once they reach that mark, they would classify at least one rival as friend in 6.9% of cases.

## 3.3 Quality model results

Similar to the analysis of the cynical model, the marginal probability for a single reviewer class in the quality model is shown in Fig 6. The results indicate that one needs to suggest a reviewer on approximately 400 submissions before they can strongly classify the reviewer in the median case.

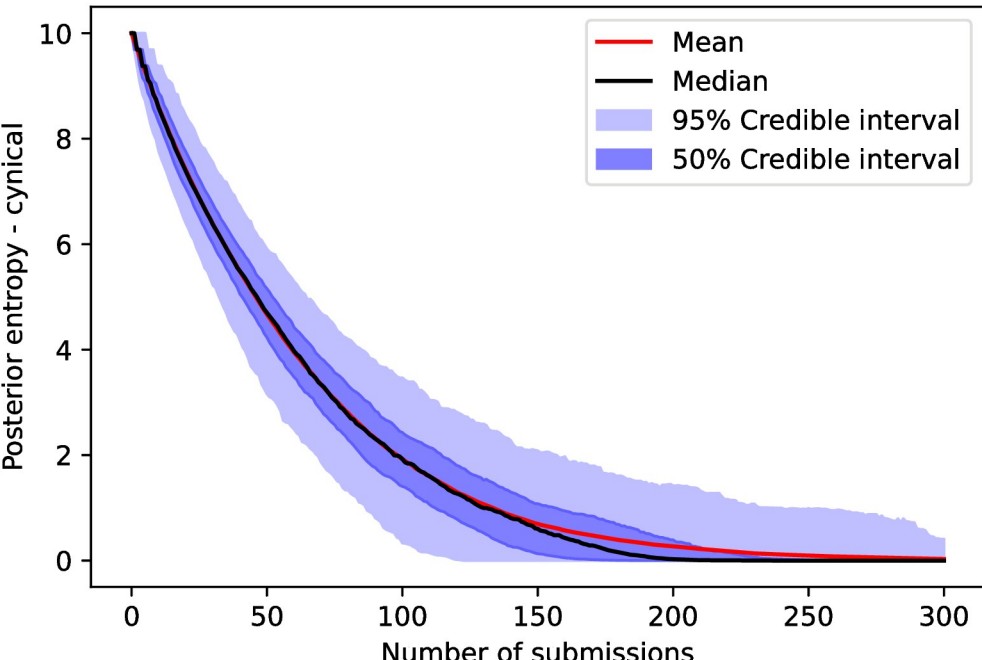

**Fig 4. The posterior's entropy—defined in (12)—as a function of the number of submissions in the cynical model.**
We observe that, in the cynical model, we need between 150 and 200 reviewed submissions in order for the entropy of a median trajectory to reach zero, meaning that for half of submissions, the posterior only fully classifies a set of 10 reviewers after 150 submissions.

MAP errors as a function of the number of submissions is shown in Fig 7 indicating that we would need more than 500 submissions to correctly classify reviewers through MAP in the median case. We would need a little less than 2000 to find the correct configuration within a .95 credible interval.

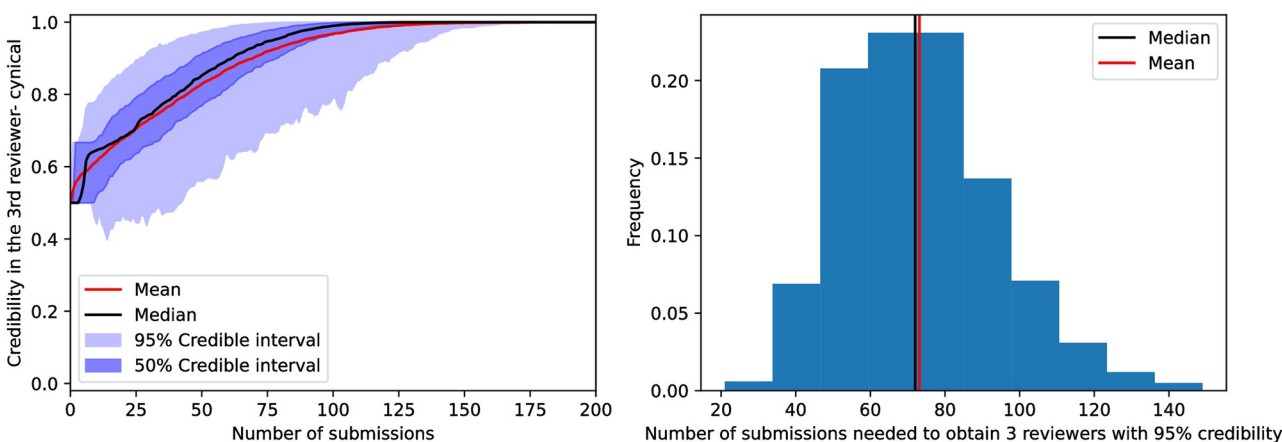

**Fig 5. The left panel presents the marginal probability of the third most likely reviewer (according to the posterior) to be friendly—defined in (13)—as a function of the number of submissions.** We observe that the median trajectory's credibility in the third reviewer reaches 95% after approximately 80 submissions in the median case. In the right panel, we see the number of submissions taken to reach 95% credibility for the same metric per sampled simulation. We observe that the mean and median number of submission is slightly bigger than 70.

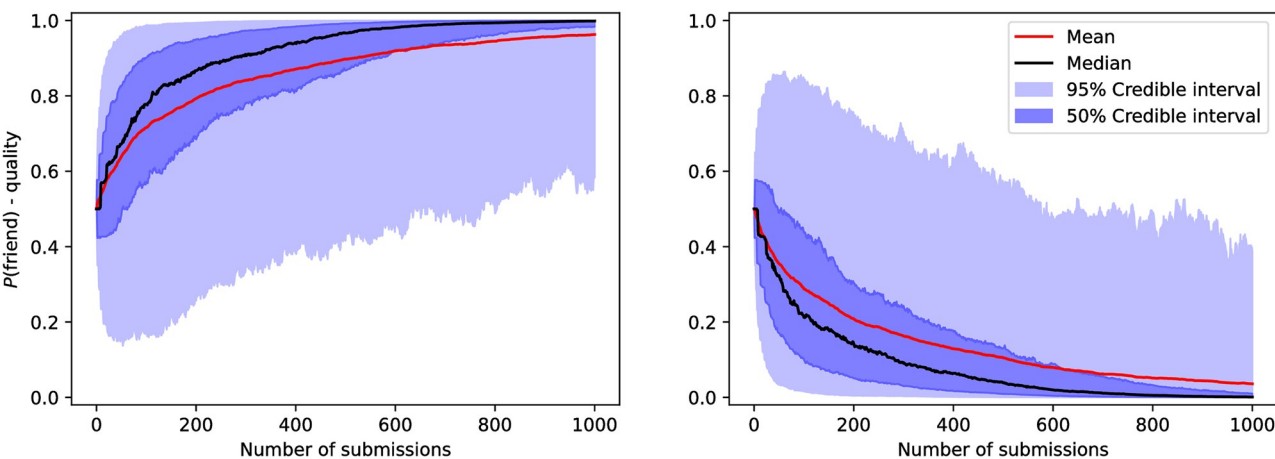

**Fig 6. Marginal posterior probability over a single reviewer's class, analogous to Fig 2, for the quality model.** We observe that the median trajectory indicates that a single reviewer ought to be suggested in approximately 400 submissions in order to reach a probability of 0.95 for the correct class.

The posterior entropy as a function of the number of submissions for the quality model is shown in Fig 8. The results suggest that we would need more than 1500 submissions to fully classify a set of 10 reviewers.

The third largest marginal posterior, as a function of the number of submissions, as well as the number of submissions necessary to reach 95% credibility are presented in Fig 9. We observe that, in the quality model, it takes approximately 400 submissions to find 3 friendly

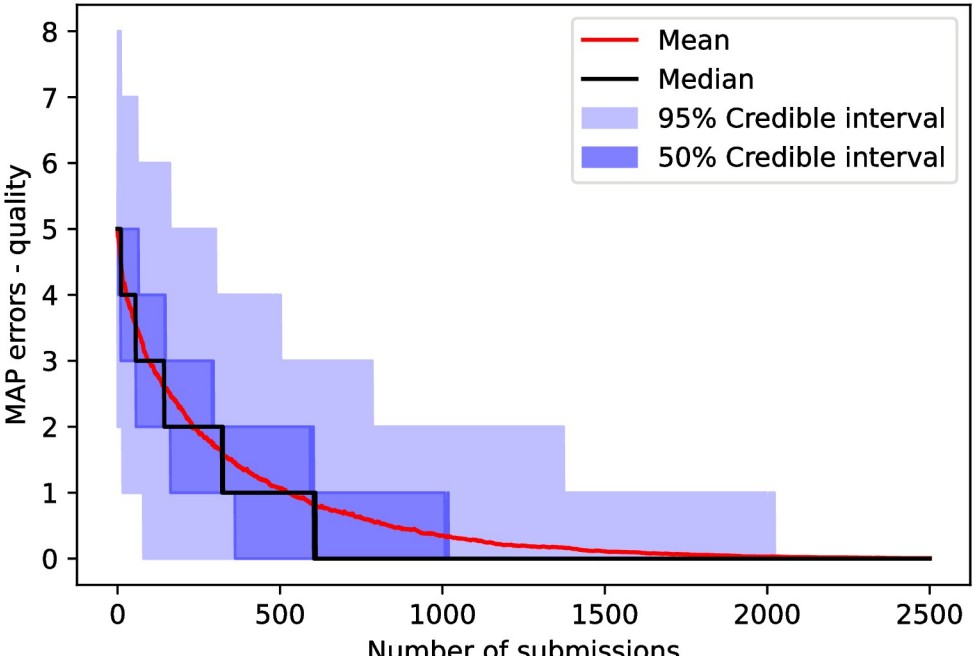

**Fig 7. The number of errors when using maximum *a posteriori* (MAP) classification, analogous to Fig 3 as a function of the number of submissions.** We observe that the median trajectory finds the correct configuration using the MAP estimate after approximately 500 submissions, while it takes around 2000 submissions to reach the correct configuration within the 95% credible interval.

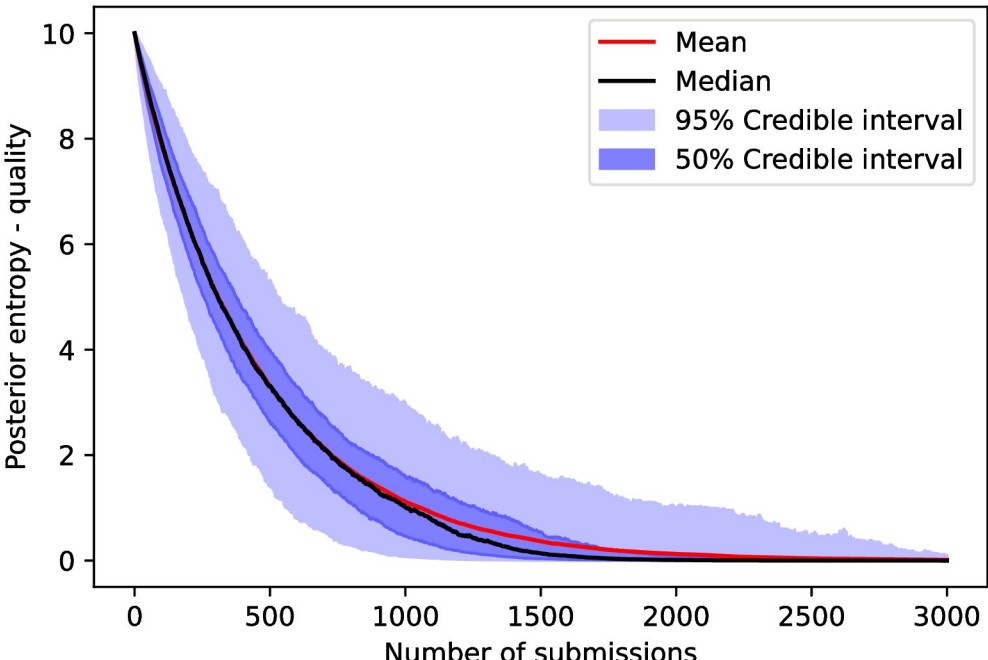

**Fig 8. The posterior's entropy, analogous to Fig 4, for the quality model.** We observe that we need around 1500 submissions in order to fully classify the reviewers (entropy approach zero) in the median trajectory.

suggested reviewers with 95% credibility. On the other hand, in the S5 File, we show that this misclassifies reviewers in less than 1.0% of datasets.

As mentioned in Sec 2.2, Figs 6–9 were constructed in a simulation where the quality factors $q_\mu$ are sampled from a Beta distribution (1) with $\alpha = \beta = 12$. We consider other sampling distributions for the quality factor and justify that this unusually tight distribution provides what is close to the overall lower bound in the S4 File. For example, any broader distribution (*e.g.*, $\alpha = \beta = 2$), only further increases the lower bound.

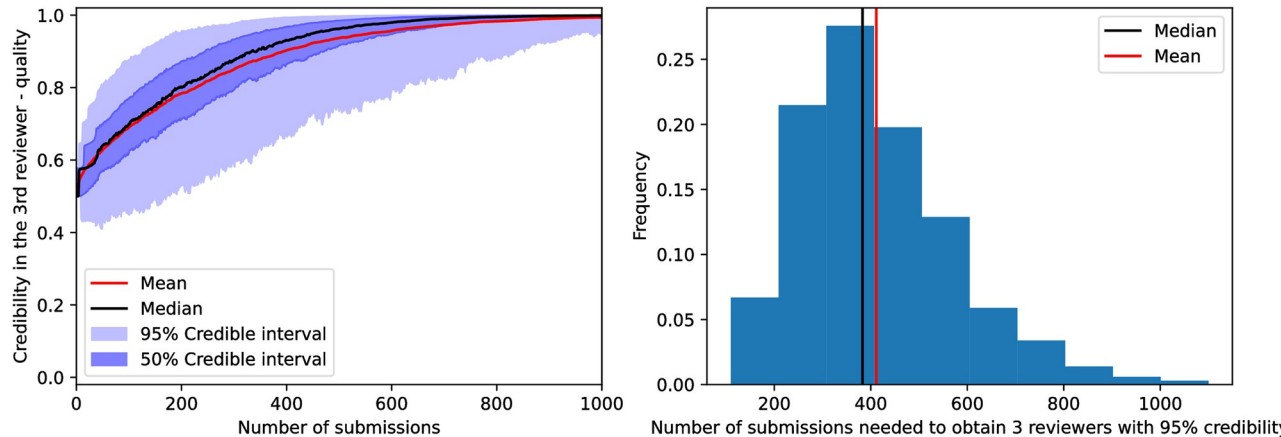

**Fig 9. The left panel presents the marginal probability of the third most likely reviewer to be friendly as a function of the number of submissions in the quality model, while the right panel presents the number of submissions taken to reach 95% credibility for the same metric, analogous to Fig 5.** Both indicate that approximately 400 submissions are necessary.

## 4 Discussion

Assessing whether a reviewer is positively or negatively inclined is a question riddled with challenges. Although we can find publicly available data on peer-review results (*e.g.* [25, 26])—allowing to quantify potential biases based on factors such as prestige [27], gender [28–30], ethnicity [31, 32], and place of origin [33, 34]—this data is anonymized to protect the privacy of authors and reviewers. On account of this, it is not possible to track the submission history of a single author across multiple publication venues to determine whether they are biasing editorial decisons through their list of suggested reviewers. Yet, the answer to the question posed by the title is not fundamentally unknowable despite the paucity of data. This is because we can simulate, and analyze, realistic outcomes based on agent-based models.

Indeed, doing so, our study shows that it is virtually unfeasible, in a single-blind peer review process, for authors to suggest reviewers that will bias the decision in their favor. Even modeling the most cynical and predictable reviewer behavior, we find that an author requires about 100 submissions to correctly classify even a set of 10 reviewers, while it takes about 70 submissions to even find 3 friendly reviewers with high credibility (see Figs 2–5). When the model is upgraded to a more realistic one (albeit still too simple), at least 400 submissions become necessary for the same task (see Figs 6–8).

This large number exceeds submissions of all but a small minority of even the most prolific scientists. Moreover, large submission numbers introduce further complications. For example, a reviewer may exhibit friendliness toward the author in one area and not another, especially problematic for prolific authors who publish across fields; it is also reasonable to expect that the reviewer may change their opinions in the time necessary to write hundreds of articles.

Further mitigating the severe idealizations of even our marginally more realistic model, would only further compound the difficulty in identifying reviewers. This would be true of any further layer of stochasticity introduced. For example: we may account for more nuanced reviewer reports, rather than classifying them as only accept or reject; allow an original pool of reviewers to change as the author gains more experience in the field; allow for neutral suggested reviewers; allow friends to become neutral or rivals over time (or vice versa); allow the author's quality factor distribution to change over time; allow the editor to select a variable number of suggested reviewers.

Another layer of stochasticity is the difference in expectations among editors and reviewers across different journals. Some journals are intended for a very specific audience, while others cater to a broad readership. Additionally, some journals prioritize scientific rigor above all else, while others prioritize publication citation rates (e.g., reviews). As a result, the same manuscript may possess different quality factors when submitted to various journals, creating uncertainty for authors attempting to determine their own quality factor distributions.

Naturally, this study assumes that the author tries to identify reviewers using only information available to them. Cases of fraud or collusion should be handled through careful editorial scrutiny. While our simulation assumes an editor that is extremely impartial, a good editor will verify if the suggested reviewers have the necessary competency to properly evaluate the submission, see *e.g.*, the Committee on Publication Ethics (COPE) guidelines [35]. Only after approved by the editor, do reviewers receive invitations. If no candidate is deemed appropriate, editors may very well select no reviewers from the suggested list introducing yet another layer of stochasticity. This is especially important in small fields where the pool of appropriate reviewers is naturally limited. Therefore the task of finding only the minimal requested number of friendly reviewers is nearly pointless for an author who publishes across fields, as is expected of prolific researchers.

Moreover, reviewers who have a close relationship with the authors may choose to withdraw voluntarily from the peer-review process upon realizing that their association could be perceived as a conflict of interest. This can limit the usefulness of the authors' suggestions.

Had a lower bound for the number of reviews found in a cynical model been small, it would have been incumbent on us to consider these complexities in order to identify which, if any, assure the soundness of the single-blind review process. But this is not the case, and the results were even surprising to us. Indeed, even the simplest model confirms that the single-blind review process is sufficiently reliable to allow authors to suggest their own reviewers without clouding or otherwise biasing the publication decision.

## Supporting information

**S1 File. Cynical model results for different numbers of suggested reviewers.**
(PDF)

**S2 File. Results with a larger ratio of friendly reviewers.**
(PDF)

**S3 File. Sampling details.**
(PDF)

**S4 File. Quality results with different parameters.**
(PDF)

**S5 File. Errors and aggressive strategy for the fourth metric.**
(PDF)

## Author Contributions

**Conceptualization:** Pedro Pessoa, Steve Pressé.

**Formal analysis:** Pedro Pessoa.

**Investigation:** Pedro Pessoa.

**Methodology:** Pedro Pessoa.

**Supervision:** Steve Pressé.

**Writing – original draft:** Pedro Pessoa, Steve Pressé.

**Writing – review & editing:** Pedro Pessoa, Steve Pressé.

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
