## [Decision Letter · Decision Letter 0]

13 Mar 2023

PONE-D-23-03685How many submissions does it take to discover friendly suggested reviewers?PLOS ONE

Dear Dr. Presse,

Thank you for submitting your manuscript to PLOS ONE. After careful consideration, we feel that it has merit but does not fully meet PLOS ONE’s publication criteria as it currently stands. Therefore, we invite you to submit a revised version of the manuscript that addresses the points raised during the review process.

We look forward to receiving your revised manuscript.

Kind regards,

Paolo Cazzaniga

Academic Editor

PLOS ONE

“This work is supported by funds from the National Institutes of Health (grant No. R01GM134426 and R01GM130745).”

“This work is supported by funds from the National Institutes of Health (https://www.nih.gov/) grant No. R01GM134426 and R01GM130745 both awarded to SP.

The funder did not play any role in the study design, data collection and analysis, decision to publish, or preparation of the manuscript.”

Additional Editor Comments:

I agree with the reviewers about the quality of the manuscript.

I therefore suggest to accept it after minor modifications, as suggested in the points raised by the reviewers.

Reviewers' comments:

Reviewer's Responses to Questions

**Comments to the Author**

1. Is the manuscript technically sound, and do the data support the conclusions?

Reviewer #1: Yes

Reviewer #2: Yes

2. Has the statistical analysis been performed appropriately and rigorously? 

Reviewer #1: Yes

Reviewer #2: I Don't Know

3. Have the authors made all data underlying the findings in their manuscript fully available?

Reviewer #1: Yes

Reviewer #2: Yes

4. Is the manuscript presented in an intelligible fashion and written in standard English?

Reviewer #1: Yes

Reviewer #2: Yes

5. Review Comments to the Author

Reviewer #1: In this work the authors combined agent-based models to emulate single-blind peer review processes and a Bayesian inference system to set a lower bound in terms of how many papers are necessary to an author to determine the "friendliness" of a reviewer. The results presented show that even in the most simple scenarios, the lower bound is too high for non very prolific authors.

The paper is well written in English.

I suggest to accept this paper after some minor revisions suggested below.

line 46, I think that the term "unfeasibly" should be changed into "unfeasible".

I don't think that a new paragraph is required at lines 84/85.

line 96, the authors forgot to put a "the" before easiest.

I don't know if such a problem is due to the submission process or intended by the authors, but in the latter case I suggest them to remove the red squares around the hyper-text links.

line 203, the authors state "for a fixed a data set", the 'a' after 'fixed' is not necessary.

I suggest to the authors to use vector images, such as pdf images, to strongly improve the readability of the figures and the overall quality of the paper.

In this work, the authors leveraged simple models and already obtained sound lower bounds. Nonetheless, I think it would be interesting to discuss how the lower bounds increase if even more complicated models are considered. I'm aware that this request can be out of the scope of the paper, but models characterized by more realistic rules and numbers can be of interest and give more realistic bounds. I suggest them to evaluate how the boundaries change if an author can suggest more than 3 reviewers and if a large number of impartial reviewers are considered; i.e., the set R is larger.

Reviewer #2: This is a very interesting and, I believe, novel exercise to ascertain how easy it is for authors to influence the publication outcome of their articles by suggesting potentially 'friendly' reviewers. Reassuringly, the answer based on the agent-based model used here is that it is extremely difficult. As the authors state, 'the single-blind review process is sufficiently reliable to allow authors to suggest their own reviewers without clouding or biasing the publication decision.’

Being a non-expert in agent-based modelling, I am not qualified to comment on the methodology used or the accuracy of the interpretation of the results. Actually this may not be too problematic, because many people for whom this work will be of interest may also lack training in ABM, and the authors might consider elaborating some concepts to help understanding. If any of my comments below betray my lack of expertise in ABM, I hope the authors will make allowance for this.

That aside, I do have a few observations and questions relating to the broader questions behind the authors’ work:

• They state that one of the challenges in exploring whether reviewers are positively or negatively inclined is that manuscript history and reviews are ‘kept under lock and key’. This is not correct; some of the studies they cite are based on actual journal data. Moreover, the PEERE group assembled a vast data set that, although now a few years old, could be used to explore the question (https://dataverse.harvard.edu/dataset.xhtml?persistentId=doi:10.7910/DVN/3IKRGI). If the authors wish to comment in their Discussion on future possible avenues of study, they could encourage others to apply some of their methods to real data sets.

• The modelling is predicated on a binary ‘accept’ – ‘reject’ inclination on the part of reviewers. In practice there is more of a spectrum from positive to negative sentiment on the part of reviewers about a manuscript. This affects the applicability of the ABM and is a limitation that I don’t believe the authors commented on, unless I missed it.

• Authors may submit the same article to multiple journals. How much would this affect the outcome of the model used by the authors? ‘Submission’ and ‘manuscript’ are often used interchangeably, but it seems important to distinguish here between the actual manuscript and the act of submission. Moreover, different journals may set different expectations for reviewers even for the same manuscript – for example, reviewers for a journal that looks for novelty may evaluate the same manuscript differently from how they would evaluate it for a journal that looks only for sound science. This is another layer of stochasticity that should be mentioned.

• Could another layer of stochasticity be that a niche field will have a much smaller pool of reviewers, and possibly also reviewers may be either much more positively inclined towards the author’s work – because they know the author well – or much more negatively inclined towards it – because it competes directly with their own?

• I am not sure I understand why, in the model, authors have access only to the number of positive reviewers, and not also to the number of negative reviews.

• How would the results of the model be affected if potentially friendly reviewers were to decline to review (because, for example, they realise they have a conflict of interest)?

6. PLOS authors have the option to publish the peer review history of their article (what does this mean?). If published, this will include your full peer review and any attached files.

Reviewer #1: **Yes: **Daniele M Papetti

Reviewer #2: No

---

## [Author Response · Author response to Decision Letter 0]

20 Mar 2023

We are thankful for the editorial efforts and reviewers suggestions in our manuscript. Overall, both reviewers have indicated agreement with publication. In the attached "Response to Reviewers" file we answer to give a point-by-point answer to each of their queries. Changes in the manuscript are marked in red.

---

## [Decision Letter · Decision Letter 1]

27 Mar 2023

How many submissions are needed to discover friendly suggested reviewers?

PONE-D-23-03685R1

Dear Dr. Presse,

We’re pleased to inform you that your manuscript has been judged scientifically suitable for publication and will be formally accepted for publication once it meets all outstanding technical requirements.

Kind regards,

Paolo Cazzaniga

Academic Editor

PLOS ONE

Additional Editor Comments (optional):

Reviewers' comments:

Reviewer's Responses to Questions

**Comments to the Author**

1. If the authors have adequately addressed your comments raised in a previous round of review and you feel that this manuscript is now acceptable for publication, you may indicate that here to bypass the “Comments to the Author” section, enter your conflict of interest statement in the “Confidential to Editor” section, and submit your "Accept" recommendation.

Reviewer #1: All comments have been addressed

Reviewer #2: All comments have been addressed

2. Is the manuscript technically sound, and do the data support the conclusions?

Reviewer #1: Yes

Reviewer #2: Yes

3. Has the statistical analysis been performed appropriately and rigorously? 

Reviewer #1: Yes

Reviewer #2: I Don't Know

4. Have the authors made all data underlying the findings in their manuscript fully available?

Reviewer #1: Yes

Reviewer #2: Yes

5. Is the manuscript presented in an intelligible fashion and written in standard English?

Reviewer #1: Yes

Reviewer #2: Yes

6. Review Comments to the Author

Reviewer #1: (No Response)

Reviewer #2: Thank you very much for responding to my suggestions and queries. I am happy to recommend the manuscript be published without any further changes.

7. PLOS authors have the option to publish the peer review history of their article (what does this mean?). If published, this will include your full peer review and any attached files.

Reviewer #1: **Yes: **Daniele M Papetti

Reviewer #2: **Yes: **Michael Willis

---

## [Editor Report · Acceptance letter]

3 Apr 2023

PONE-D-23-03685R1 

How many submissions are needed to discover friendly suggested reviewers? 

Dear Dr. Pressé:

I'm pleased to inform you that your manuscript has been deemed suitable for publication in PLOS ONE. Congratulations! Your manuscript is now with our production department. 

Kind regards, 

on behalf of

Dr. Paolo Cazzaniga 

Academic Editor

PLOS ONE